# Phenolic Acids and Flavonoids in Acetonic Extract from Quince (*Cydonia oblonga* Mill.): Nutraceuticals with Antioxidant and Anti-Inflammatory Potential

**DOI:** 10.3390/molecules27082462

**Published:** 2022-04-11

**Authors:** Karen Marlenne Herrera-Rocha, Nuria Elizabeth Rocha-Guzmán, José Alberto Gallegos-Infante, Rubén Francisco González-Laredo, Mar Larrosa-Pérez, Martha Rocío Moreno-Jiménez

**Affiliations:** 1Research Group on Functional Foods and Nutraceuticals, TecNM/Instituto Tecnológico de Durango, Felipe Pescador 1830 Ote., Durango 34080, Mexico; 16041429@itdurango.edu.mx (K.M.H.-R.); agallegos@itdurango.edu.mx (J.A.G.-I.); gonzalezlaredo@gmail.com (R.F.G.-L.); 2Department of Nutrition and Food Science, Facultad de Farmacia, Universidad Complutense de Madrid, 28040 Madrid, Spain; mlarrosa@ucm.es

**Keywords:** quince, phenolic compounds, acetonic extracts, nutraceuticals

## Abstract

Quince (*Cydonia oblonga* Mill.) is a potential source of polyphenolic compounds related with beneficial biological processes. In this study polyphenols from quince fruit were extracted with aqueous acetone at different ratios. A polyphenol profile was identified and quantified by LC-ESI-QqQ. The antioxidant capacity (ORAC and DPPH) and anti-inflammatory effect (inhibition of COX-2 cyclooxygenase) were evaluated in vitro. The results indicated an effect of the aqueous acetone ratio on the extraction of polyphenolic compounds. The higher extraction yields of polyphenolic compounds were attained with 60–75% aqueous acetone. However, extracts obtained with 85% aqueous acetone promoted higher antioxidant and anti-inflammatory effects. Optimal scaling analysis indicated that hydroxycinnamic acids (quinic and chlorogenic), hydroxybenzoic acids (vanillic and syringic), flavonoids (quercetin and kaempferol), dihydrochalcones (neohesperidin) and flavones (acacetin) are related to the antioxidant activity of quince. While phenolic acids, flavonols (kaempferol-3-O-glucoside and rutin) and flavanols (epicatechin) generated the anti-inflammatory effect by inhibiting 52.3% of the COX-2 enzyme. Therefore, a selective extraction of phenolic mix can reduce oxidative stress or inflammatory processes. This suggests the use of quince as a natural source with significant nutraceutical potential.

## 1. Introduction

The importance of an adequate diet to maintain health is now evident. These days the population is focusing more on disease prevention rather than in treatment. Epidemiological studies indicate that oxidative stress and inflammatory processes represent a high risk for the development of chronic diseases [1]. In this sense, the use of products that allow the prevention or treatment of various pathologies has been chosen, among these are nutraceuticals, which are concentrated formulas of bioactive compounds or secondary metabolites obtained from food matrices [2,3]. Polyphenolic compounds are secondary metabolites of plants with significant bioactivity that is associated with antioxidant, anti-inflammatory, cardioprotective, and anticancer effects, among others [4,5]. Consequently, these bioactive compounds may be used for the formulation of nutraceuticals.

*Cydonia oblonga* Mill. (Quince) is a plant of the Rosaceae family, used for nutritional and medicinal purposes due to its high content of polyphenols [6]. The quince fruit is highly perishable and has a distinctive astringent flavor; therefore, its consumption as fresh fruit is in small proportion and instead, jelly, liquor, and dried preserves are produced [7]. In industry, the fruit is used as a source of pectins for the treatment of inflammatory bowel diseases [8]. Traditionally, all parts of the plant are used to treat various pathologies: the seeds of the plant are used to treat diarrhea, cough, dysentery, sore throat, constipation, and bronchitis [9]. Leaf extracts reduce diabetes, cancer, and hyperlipidemia [10,11]. The fruit has been also used to treat sore throats, constipation, and bronchitis [12]. Authors attribute these effects to various phenolic compounds such as caffeoylquinic acid, quinic acid, shikimic acid, rutin, kaempferol-3-O-glucoside, quercetin-3-O-galactoside, kaempferol-3-O-rutinoside, as well as flavanols [13,14]. In recent studies, extractions of leaves, stems and fruits from quince were carried out using ultrasound assistance and methanol as solvent. They established the nutraceutical potential of quince with antioxidant capacity and the inhibition of digestive enzymes by crude extracts, which were rich in phenolic acids, flavonoids, lignans, and stilbenes [15]. Other organic solvents such as acetone have been widely used in the pharmaceutical industry for the separation of fractions abundant in bioactives from herbs, roots, and flowers. Studies suggest the use of aqueous acetone for the selective extraction of flavonoids (i.e., quercetin, quercetin glucosides and rutin) related to antioxidant and anti-inflammatory activities, establishing that the extraction of these compounds and their biological effects depend on the ratio of aqueous acetone used [16].

However, the use of organic solvents in fruits rich in polysaccharides involves several problems in two main ways: interactions of polyphenols with methyl groups from pectin by non-covalent bonds and by the covalent bonds formed between nucleophilic molecules and quinones from polyphenols [17,18]. The latter are important, because the polyphenols could be bound and insoluble, reducing their extraction yield. This behavior is important in fruits rich in high-methoxyl pectins and with significant presence of flavan-3-ols and their polymers (procyanidins), specially with high degree of polymerization (DP), as in *Cydonia oblonga* Mill. Several reports indicate higher amount of pectins in quince compared to apples [19]; the degree of methylation indicates the presence of high-methoxyl pectins, which limits the polyphenol extraction by organic solvents. Therefore, the objective of this work was to identify the solvent ratios of aqueous acetone, which allow the extraction of antioxidant and anti-inflammatory bioactives with potential use for the formulation of nutraceuticals from the fruit of *Cydonia oblonga* Mill.

## 2. Material and Methods

### 2.1. Chemical Reagents

Pepsin, pancreatin, α-amylase, amyloglucosidase, standards of gallocatechin, catechin, epicatechin, procyanidin B2, quercetin, quercetin 3-O glucoside, naringenin, naringin, luteolin, apigenin, acacetin, rutin, neohesperidin, taxifolin, phloretin, kaempferol, kaempferol-3-O-glucoside, quinic acid, protocatechuic acid, 2,5-dihydroxybenzoic acid, 4-hydroxybenzoic acid, 2,4,6-trihydroxybenzaldehyde, syringic acid, chlorogenic acid, 4-O-caffeoylquinic acid, caffeic acid, coumaric acid, formic acid, and shikimic acid were obtained from Sigma Chemical (St. Louis, MO, USA). Acetone, methanol and acetonitrile were LC-MS grade from J.T. Baker Inc. 6-hydroxy-2,5,7,8-tetramethylchroman-2-carboxylic acid (TROLOX, Sigma-Aldrich, St. Louis, MO, USA), 2,2′ -azo-bis (2-amidino-propane) dihydrochloride (AAPH, Sigma-Aldrich, USA), 2,2-diphenyl-1-picrylhydrazyl (DPPH), COX-2 kit no. 560,131 from Cayman Chemical Co., Ann Arbor, MI, USA.

### 2.2. Pulp of Quince Fruit

The quince fruit (*Cydonia oblonga* Mill.) is a cultivated species and unique in Northern Mexico. An herbarium specimen, voucher for the samples used in this study was collected by MR Moreno-Jimenez and identified by Dr. Socorro González-Elizondo, from the CIIDIR herbarium [20], where it was deposited with the accession number 58696. The botanical sample and the fruits used for this experiment were acquired directly from the producer site at El Salto, Pueblo Nuevo, Durango, México (Latitude: 23.7794, Longitude: −105.362) from the September 2019 season. Quince pulp was obtained from whole mature fruit. The quince was sanitized and processed in an industrial pulper (Polinox, Cd de México, México) at Instituto Tecnológico de Durango, México. The pulp obtained was stored at −20 °C and subsequently lyophilized (0.045 mBar, −51 °C) (FreeZone 18 lyophilizer, LABCONCO, Kansas City, MO, USA) for further processing and analysis.

### 2.3. Determination of Dietary Fiber and Fiber-Bound Polyphenols

Determination of dietary fiber was developed according to Goñi method [21] with some modifications. A sample (300 mg) of lyophilized quince was weighed and resuspended in phosphate buffer (pH 1.5). Subsequently, 0.2 mL of pepsin solution was added and incubated at 40 °C for 1 h. The pH was adjusted to 7.5 with a NaOH solution, then 1 mL of pancreatin solution was added and incubated at 37 °C for 6 h. Buffer (10 mL) was added again and adjusted to pH of 6.9. An α-amylase solution was added (1 mL) and incubated for 16 h at 37 °C with continuous shaking. Samples were centrifuged at 3000 g for 15 min and the supernatants were removed. The pellets were washed two times with distilled water, dried overnight at 105 °C, placed in a desiccator, and subsequently weighed for determination of the total content of insoluble fiber. Later, 10 mL of sodium acetate buffer (0.2 M) at pH 4.75 was added to the supernatants, plus 0.1 mL of amyloglucosidase, and subsequently incubated at 60 °C for 45 min. The mixture obtained were dialyzed (Spectrum™ Spectra/Por™ 4 RC Dialysis Membrane Tubing 12,000–14,000 Dalton MWCO) with water exchange every 24 h for 48 h at 37 °C. Weight of collected samples was determined, defining the soluble fiber. Determination of polyphenols bound to the fiber was carried out by means of a liquid–liquid extraction with ethyl acetate to the obtained sample, which was brought to dryness (Labconco centrifugal concentrator, Kansas City, MO, USA). Aliquots were saved for later analysis.

### 2.4. Obtaining Crude Extracts

Mixtures of aqueous acetone in ranges from 60 to 100% were used to obtain crude extracts. Lyophilized quince pulp was homogenized at 1:10 *w/v* ratio and 24,000 rpm, using a disperser (IKA^®^ T10 ULTRA-TURRAX, Darmstadt, Germany). Suspensions were then clarified by centrifugation (Labofuge Heraeus 400R centrifuge, Asheville, NC, USA) at 10,000 rpm for 10 min at 4 °C. Extracts were aliquoted in volumes of 2 mL and subsequently the solvent was evaporated (Labconco centrifugal concentrator, Kansas City, MO, USA). Dry aliquots were reserved for later analysis.

### 2.5. Analysis of Soluble Polyphenols and Crude Extracts by UPLC-PAD/ESI-QqQ MS/MS

The samples were analyzed for the identification and quantification of phenolic compounds according to the Díaz-Rivas method [22]. The samples were resuspended in 200 µL of methanol and filtered using 0.45 µm PTFE filters before analysis. The UPLC system consisted of a quaternary bomb (QSM—“quaternary solvent module”) and a sample handler (SM-FTN) Acquity class-H (Waters Corp., Milford, MA, USA) coupled with a tandem Xevo TQ-S triple quadrupole mass spectrometer (Waters Corp., USA). Data were recorded in a multiple reaction monitoring mode (MRM). Data acquisition and processing were performed using MassLinx software (Waters Corp., USA).

Chromatographic separations were performed on a C18 Acquity UPLC BEH column (100 mm × 2.1 mm × 1.7 µm) (Waters Corp. USA) operated at 35 °C, using 7.5 mM water/formic acid (A) and acetonitrile (B) as mobile phases at 0.35 µL/min, and sample volume of 2 µL. A solvent gradient was applied starting with 3% B, maintaining flow for 1.23 min, followed at 9% B to 3.82 min, followed at 16% B to 11.40 min, followed at 50% B to 13.24 min, followed at 3.0% to 15 min, returning to the initial conditions (3% B). A negative ionization was used for the MS test. ESI conditions were followed: capillary voltage, 2.5 kV; desolvation temperature, 300 °C; temperature source, 150 °C; desolvation and cone gas, 500 L/h and 151 L/h, respectively, and collision gas, 0.13 mL/min. MRM transitions were determined by MS/MS spectrum of existing phenolic acid standards and a mixture of different phenolic compounds was used as a monitor for retention time and m/z values. The identification of the peaks was based on the comparison of their retention times and MRM transitions with those of pure standards. Quantitative determinations of phenolic compounds were carried out using calibration curves of the available standards from Sigma Chemical (St. Louis, MO, USA) (Table 1).

### 2.6. Antioxidant Assays

#### 2.6.1. Oxygen Radical Absorbance Capacity (ORAC)

ORAC assays were performed on different extracts as described by Ou et al. [23]. The assay was conducted in 7.5-mM phosphate buffer (pH 7.4) at 37 °C. Fluorescein was used as fluorescent probe, Trolox as positive control (0–50 mM) and AAPH, as a peroxyl radical generator. The samples (20 µL) were mixed with 180 µL of fluorescein (0.108 µM) and the plates pre-incubated at 37 °C for 15 min (Synergy^TM^ HT Multi-Detection Microplate Reader, Bio-Tek, Winooski, VT, USA). After this pre-incubation period, 75 μL of AAPH (79.65 mM) were added to the reaction. Subsequently, the fluorescence was recorded over a period of 150 min, with data captured every 210 s at a wavelength for excitation of 485 nm and for emission of 580 nm. The final ORAC values were determined using the area under the curve and expressed as µM Trolox equivalents per mL.

#### 2.6.2. DPPH Assay

DPPH assay was developed according to Brand-Williams method [24]. The 2,2-diphenyl-1-picrylhydrazyl radical was dissolved at a concentration of 100 µM in 80% methanol, protected from light and used immediately after being prepared. Subsequently, in a 96-well microplate, 6 µL (100 µg) of sample of each of the percentages of aqueous acetone or Trolox as positive control and 194 µL of DPPH radical were added. The mixture was incubated at room temperature for 30 min in the dark. After that, the absorbance of the reaction products was taken (Synergy^TM^ HT Multi-Detection Microplate Reader, Bio-Tek, Winooski, VT, USA) at 515 nm. Results were expressed as %RSA (radical scavenging activity)/µM Trolox equivalents per mL, considering a standard curve of Trolox (25–100 mM, R^2^ = 0.9786). The parameter that was measured is the percentage reduction of DPPH (% RSA) compared to the sample, Equation (1). By linear regression, curves of the percentage of inhibition vs. concentration of the solutions of obtained by the next linear regression equation Trolox (µM/µg):% Inhibition = [([Abs]_control_ − [Abs]_sample_)/[Abs]_control_] × 100(1)

### 2.7. Anti-Inflammatory Assay

An enzyme inhibition detection of cyclooxygenase-2 was carried out using a commercial COX-2 inhibitor screening kit (fluorometric), following manufacturer′s specifications (Cayman Chemical CO, Ann Arbor, MI, USA). Aqueous acetone extracts obtained (10 μL), lyophilized prostaglandin (positive control) and COX-2 inhibitor DuP-697 (60 μM) were added to reaction tubes with 160 μL of buffer, 10 μL of heme prosthetic group and 10 μL of COX-2 enzyme. The reaction solution was incubated for 10 min at 37 °C, then 10 µL of arachidonic acid was added, mixed rapidly, and incubated for 2 min at 37 °C. The reaction was terminated by adding 30 µL of tin chloride, then removed from the heat and incubated for 5 min at room temperature. The prostaglandins were quantified by ELISA, 50 μL of each sample was added in duplicate to each well, in addition to 50 μL of AChE traces and 50 μL of ELISA antiserum. Then, it was incubated for 18 h at room temperature on an orbital shaker. At the end of incubation time, it was washed five times with ELISA wash buffer and added with 200 µL of Ellman′s reagent. The wells were covered and incubated at room temperature protecting the plate from light until reaching ≥0.3 AU when read at 420 nm (Synergy HT Multi-Detection Microplate Reader Spectrophotometer, Bio-Tek, Winooski, VT, USA). The results were managed using manufacturer’s resources (Elissa Double worksheet, www.caymanchem.com/analysis/eia, accessed on 15 July 2021) and were expressed as an inhibition percentage (%).

### 2.8. Statistical Analysis

All results were expressed as the mean ± standard deviation. Data were analyzed by one-way ANOVA (*p* < 0.05). An optimal scaling analysis was performed for the categorical comparison between variables. Statistical analyses were performed using IBM SPSS Statistics 22.0 software (IBM Corp., Endicott, NY, USA).

## 3. Results and Discussion

### 3.1. Polyphenols Extracted from Quince Pulp (Cydonia oblonga Mill.) with Aqueous Acetone

The results obtained from the fiber content assay indicated that the quince pulp has an abundant content of dietary fiber (820.50 ± 0.70 mg/100 g) and most of the polyphenol compounds are bound to the pulp (30,218 ± 104 µg/g of freeze-dried pulp), identifying mainly flavanols, hydroxycinnamic acids, hydroxybenzoic acids, flavanols and to a lesser extent dihydrochalcones (Figure 1). However, extractable (i.e., soluble) polyphenols are of great importance, and unlike bound polyphenols, usually, they can be extracted with different polar and moderately polar solvents. Several reports in fruits such as apples indicate that the presence of high contents of flavan-3-ol are linked to pectins, and these polymers make the proper extraction of polyphenols difficult [25]. However, there are opposite reports on the use of solvents such as acetone in different acetone–water ratios that allow the selective extraction of higher contents of polyphenolic compounds with biological potential [26]. Therefore, in this study the extraction was carried out with different ratios of aqueous acetone and the influence on the extraction profile and yields of phenolic compounds in quince pulp were determined. The results showed the presence of 12 flavonoids and 15 phenolic acids, which were identified through the analysis by UPLC-PAD/ESI-QqQ MS/MS determining their retention times, molecular weights (*m*/*z*), main ion transitions, and λ max (Table 1). A similar polyphenol profile was observed in all aqueous acetone extracts at different concentrations, i.e., mainly hydroxycinnamic acids, followed by flavonols, flavanols, hydroxybenzoic acids, and dihydrochalcones (Figure 2).

In general, it was determined that the acetone concentration affects the yield of extracted polyphenol compounds, extracting the higher contents of polyphenols at 60–75% acetone and the lower extraction yield at 90–100% (Figure 3). These results are related to the polarity of acetone, the water solubility of polyphenol compounds in acetone mixtures, their chemical structure, and the disposition of their functional groups. Unlike the extraction with aqueous acetone, the methanolic extractions of the quince fruit contain high content of hydroxycinnamic acids (shikimic acid, quinic acid and their derivatives), flavonoids, and proanthocyanidins [27]. Meanwhile, the use of ultrasound assistance and methanol as extraction solvent, enhances the content of flavonoids, phenolic acids, and lignans [28].

The extraction of hydroxycinnamic acids and hydroxybenzoic acids was identified (Table 2). Quinic, chlorogenic, and caffeic acids were the main hydroxycinnamic acids extracted at all acetone ratios, with 75% acetone having the maximum contents of 562.4 ± 27.3 mg/g, 115.14 ± 15.3 mg/g, and 0.19 ± 0.0 mg/g, respectively. The extraction of phenolic acids and their abundance at the different ratios of aqueous acetone are related to the presence of hydroxyl groups on the chemical structures, allowing the formation of acetone complexes, and facilitating the extraction [29]. A variety of hydroxybenzoic acids were identified but at reduced abundance, i.e., mainly shikimic acid and derivatives of hydroxybenzoic acid. A similar behavior was reported in the literature, it was established that the quince fruit contains a lower content of hydroxybenzoic acids compared to the peels and the leaves [7]. Different subclasses of flavonoids were extracted, the flavonols were obtained in greater abundance with 60% acetone, such as rutin (281.9 ± 29.7µg), kaempferol-3-O-glucoside (4.4 ± 0.2 µg) and quercetin (0.2 ± 0.0 µg), while luteolin was detected as traces (Table 2). The chemical structure, the presence and position of hydroxyl groups, the molecular weight, and the length of hydrocarbon chains, allowed greater solubility and therefore greater extraction yields. In this regard, it has been described that acetone breaks interflavanic bonds, while water, a solvent of a polar nature, likely binds polar compounds such as flavonols, increasing their extraction performance [30].

In the same way as flavonols, flavanols as procyanidin B, epicatechin and catechin were extracted at all acetone ratios, the greater abundance was extracted with 60% acetone, i.e., mainly procyanidin B2 (259.5 ± 40.0 µg), epicatechin (3.9 ± 0.6 µg), and catechin (0.19 ± 0.06 µg) (Table 2). However, it is clear to observe lower amount of flavanols and hydroxycinnamic acids when acetone ratio was increased (95–100%), this behavior could be explained by the presence of pectic polysaccharides in quince. The nature of this compound in quince is rich in arabinans [31], increasing the interactions between pectin and polyphenols, in particular with procyanidins [32]. This interaction is the function of the procyanidin monomers (i.e., high degree of polymerization producing more strong interactions) and the pectin nature (i.e., high content of methoxyl pectins producing more strong interactions) [33]. In this respect, the degree of polymerization of procyanidins from quince was very high (DP, 16–19) in comparison with the procyanidins from apple (DP, 2–4); the pectins from quince have been reported to be rich in arabinans and with high content of methoxyl pectin type (75–92%) [34]. Under these conditions, higher level of interactions between pectins and procyanidin are expected. However, the main effect was the ratio of acetone, because at higher acetone concentration a lower presence of procyanidin was observed. It is important to note that during quince pressing to obtain pulp, polyphenols (i.e., procyanidins) were transferred from fruit vacuoles to the whole pulp [35], increasing their oxidation by polyphenol oxidase (PPO) [18]. It has been reported that procyanidins are not substrates of PPO enzyme [36]; however, it has been indicated that they can be converted into *o-*quinones and reduced back to their *o*-diphenol form [37]. This reaction produces covalent bonds between procyanidins and pectins. This fact reduces the extraction of procyanidins, which is reinforced also by higher acetone proportions in the acetone/water mixtures. Higher concentrations of acetone (i.e., with lower dielectric constant in comparison to water) enhance the availability of procyanidins to reduction reactions, increasing covalent bonds and reducing their extraction yields [38].

Finally, a low concentration of dihydrochalcones mainly neohesperidin was found at the highest content (0.33 ± 0.03 µg/g) with 75% acetone (Table 2). The low extraction yield of dihydrochalcones is related to their distribution in the quince plant because, it is found mainly in the leaves, having a relative abundance of 3-hydroxyphloretin-2-O-xylosyl-glucoside and dihydroquercetin 3-O-rhamnoside [15].

### 3.2. In Vitro Antioxidant Effect of Polyphenols from Quince Fruit (Cydonia oblonga Mill.)

The results obtained of antioxidant capacity are shown in Table 3. In general, the antioxidant effect is related directly to the total phenolic content. The ORAC assay indicated statistically significant differences at aqueous acetone ratios. However, ratio of 85% generated higher antioxidant capacity indicated as Trolox equivalents. Different polyphenol concentrations were identified in this fraction from aqueous acetone. The major difference was identified at hydroxycinnamic acids (chlorogenic and caffeic) and 2,5-dihydroxybenzoic acid contents (Table 2). This indicated that higher concentrations of polyphenols are not related with better antioxidant capacity. In the DPPH radical scavenger assay, no statistically significant differences were observed in the aqueous acetone fractions. Probably this result is related to the diversity of polyphenolic compounds extracted at different ratios of acetone that even at low concentrations are able to stabilize the DPPH radical. Compared to Trolox (positive control), the RSA% depends on the concentration used in the assay. Therefore, it may be necessary to confirm the antioxidant capacity with additional techniques as ABTS, FRAP, or LDL. 

To obtain specifically information of correlation on the polyphenol compounds extracted with acetone (60–100%) and their biological capacities, an optimal scaling analysis was performed (Figure 4). This includes the concentrations of extracted polyphenol compounds (60–100%) and the biological response obtained. Results of the antioxidant effects showed that hydroxycinnamic acids (quinic, caffeic and chlorogenic acids), hydroxybenzoic acids (vanillic, syringic, protocatechuic and 2,5-dihydroxybenzoic acids), flavonols (kaempferol and quercetin), flavones (acacetin), and dihydrochalcones (neohesperidin) present in quince pulp were the main compounds related to antioxidant activity by the ORAC test, while no specific compound was related by the DPPH assay (Figure 4a,b). Different authors indicated that hydroxycinnamic, quinic, and chlorogenic acids have been described with significant antioxidant effect in vitro, promoting a greater scavenging of free radicals due to their structural characteristics such as the substitution of aromatic rings, the number and position of hydroxyl groups in relation to the carboxyl group, as well as the position and nature of esterification [39]. Vanillic acid, a hydroxybenzoic acid, displays antioxidant effect in vitro, eliminating free radicals and inhibiting lipid peroxidation. In in vivo models, vanillic and syringic acids reduce oxidative stress activating the AkT pathway, which promotes expression of antioxidant enzymes (i.e., SOD, Catalase and glutathione peroxidase) [40,41]. Unlike phenolic acids, a lower number of flavonoids were correlated with antioxidant activity, among them, flavonoids such as quercetin and kaempferol; these polyphenols have been widely described as powerful antioxidants in in vitro and in vivo models, modulating the scavenging of free radicals (ROS) and activation of antioxidant enzymes [42,43]. Acacetin and neohesperidin were extracted in lower concentrations; however, although they are correlated with antioxidant effects, several authors have indicated that their biological effect is not related to higher concentrations. The administration of acacetin at low doses in in vivo models activates antioxidant enzymes [44], while neohespiridin shows free radical scavenging capacity [45].

### 3.3. In Vitro Anti-Inflammatory Effect of Polyphenols from Quince Fruit (Cydonia oblonga Mill.)

Cyclooxygenase-2 (COX-2) is an inducible enzyme synthesized from arachidonic acid, responsible for producing prostaglandins. The expression of this enzyme is produced by the presence of lipopolysaccharides and proinflammatory cytokines and has been considered as a biomarker of proinflammatory processes. Compounds such as polyphenols can reduce COX-2 expression [46]. The polyphenols present in quince stimulated the inhibition of the COX-2 enzyme and the percentage of enzyme inhibition remained in a range of 15–52.3% (Table 4). The extract with 85% aqueous acetone generated the higher inhibition of the enzyme COX-2 (52.31 ± 0.01%). Although extract does not have a high concentration of polyphenols, it does contain compounds related to anti-inflammatory effects, among them are hydroxycinnamic acids, flavanols and traces of flavanone, and dihydrochalcones (Table 2). The results were compared with DuP-69, a member of the diaryl heterocycle group of selective COX 2 inhibitors, provided by the kit assay. This inhibitor generated 80% inhibition at a concentration of 60 µM.

According to the optimal scaling analysis, phenolic acids such as syringic, protocatechuic, 4-hydroxybenzoic, chlorogenic, vanillic, caffeic and syringic acids show correlation with the COX-2 inhibition in vitro (Figure 4a). Polyphenols have been reported having a similar effect as non-steroidal anti-inflammatory drugs (NSAIDs). These inhibit pro-inflammatory mediators such as COX enzymes, reducing their activity and genetic expression, in addition to inhibiting the activity of transcription factors such as nuclear factor-kβ (NF-kβ) in inflammatory and antioxidant pathways [47]. Different studies have described the anti-inflammatory effect of phenolic acids in in vivo and in vitro models. Among them, coumaric acid causes COX-2 inhibition in colon cancer models [48], while syringic and vanillic acids, in addition to inhibiting COX-2, decrease the expression of pro-inflammatory proteins such as TNFα [42]. In this sense, hydroxycinnamic acids such as chlorogenic and caffeic acids have shown an anti-inflammatory effect by reducing the expression of COX-2 in in vitro and in vivo models [49,50]. However, several authors have related these compounds primarily with the activation of pathways related to the antioxidant effect [51]. On the other hand, flavonols and flavanols such as kaempferol-3-O-glucoside, rutin, and epicatechin gallate have shown a direct correlation in the inhibition of COX-2 (Figure 4b). Therein, several authors have described these flavonoids as promising anti-inflammatory compounds by the inhibition of COX-2, as well the secretion of prostaglandins and pro-inflammatory proteins in in vivo models related to neuronal diseases [52,53]. Procyanidin B2 was the flavan-3-ol extracted in higher concentrations, but it did not show any anti-inflammatory effect according to the optimal scale analysis, although it has been reported with capacity of inhibiting cyclooxygenase in cell models [54]. However, dimer B2 should not be related as an important COX-2 inhibitor because it has been described that the anti-inflammatory and antioxidant effects of procyanidins occur mainly after they are metabolized by the intestinal microbiota and not by the molecules originally ingested [54].

## 4. Conclusions

This study has highlighted the use of aqueous acetone extracts in the selective extraction of polyphenolic compounds of biological interest from quince fruit (*Cydonia oblonga* Mill.). The extract obtained with 85% aqueous acetone resulted with the best biological impact in antioxidant and anti-inflammatory activities, being more related to the synergistic effect of the diversity of polyphenol compounds and not to their higher concentrations. The phenolic acids, flavonols, flavones, and dihydrochalcones were related to the antioxidant activity, while the anti-inflammatory effect was related to phenolic acids, flavonols, and flavanols compounds. This indicates that the selective extraction of polyphenols from the *Cydonia oblonga* fruit can be carried out for the generation of nutraceuticals with antioxidant and anti-inflammatory properties.

## Figures and Tables

**Figure 1 molecules-27-02462-f001:**
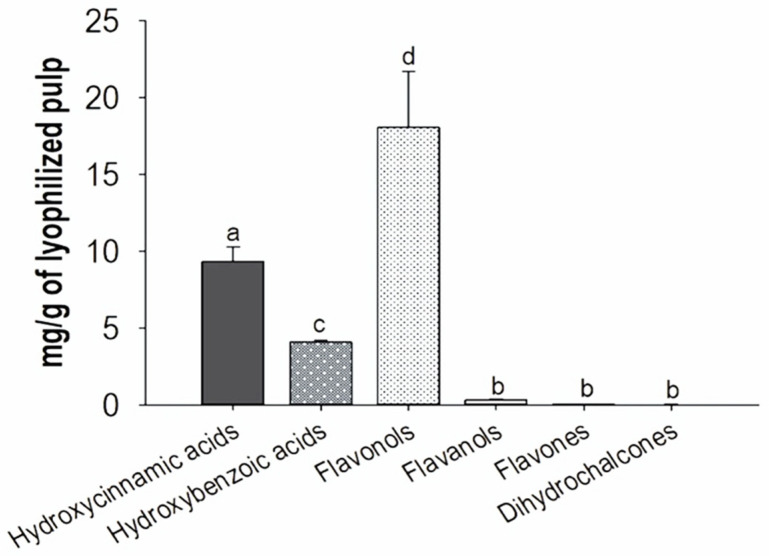
Polyphenols compounds associated with dietary fiber of Quince fruit (*Cydonia oblonga* Mill.).

**Figure 2 molecules-27-02462-f002:**
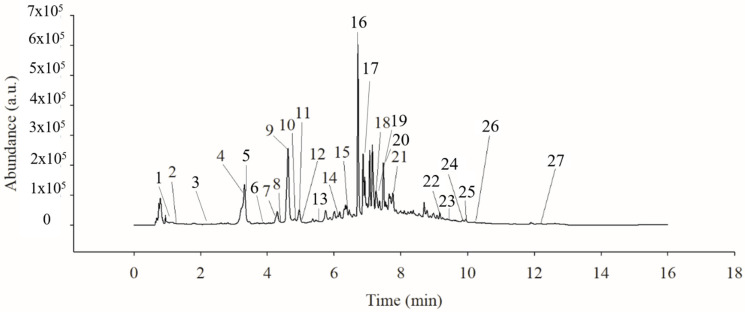
UPLC-PDA chromatogram of Quince fruit (*Cydonia oblonga* Mill.) recorded at 280 nm. Peak numbers correspond to the compounds listed in Table 1.

**Figure 3 molecules-27-02462-f003:**
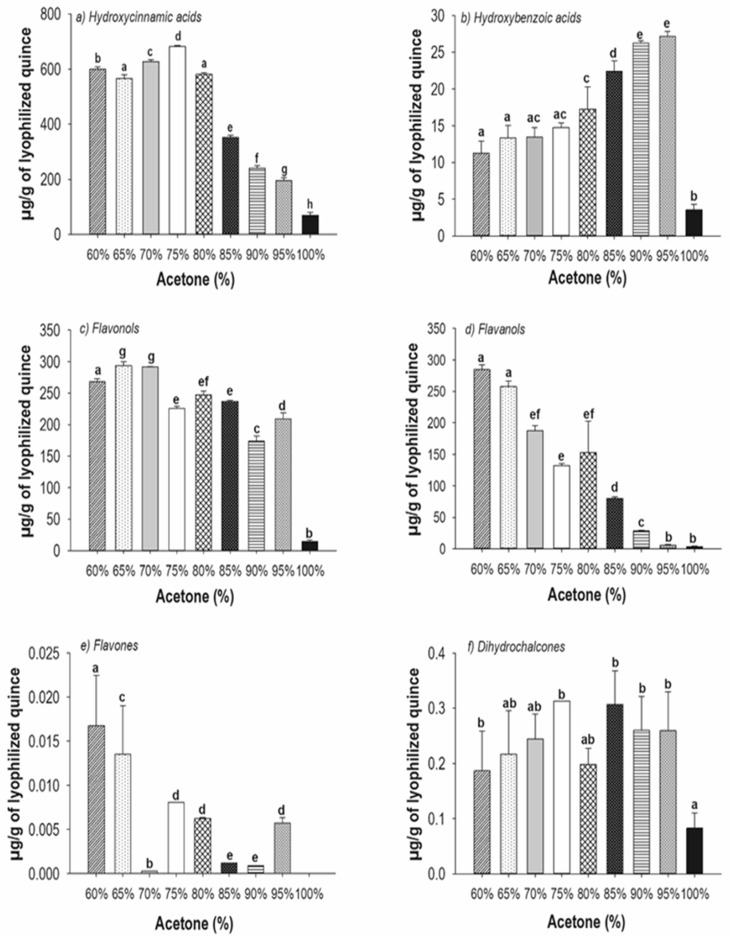
Effect of aqueous acetone on the extraction of polyphenolic compounds in Quince fruit (*Cydonia oblonga* Mill.). Different letters indicate statistical significance (ANOVA, Tukey *p* < 0.001).

**Figure 4 molecules-27-02462-f004:**
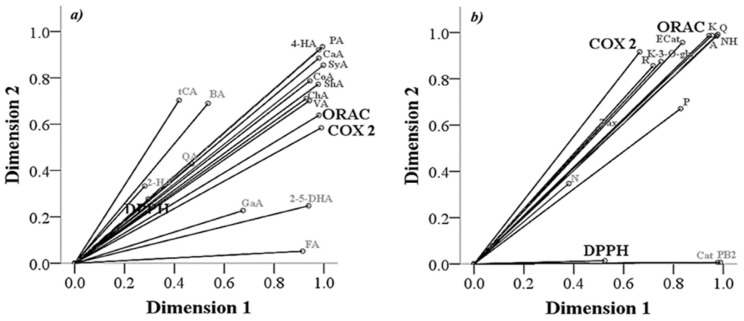
Optimal scaling analysis of polyphenols from Quince fruit (*Cydonia oblonga* Mill.) extract. Relationship between polyphenols and in vitro biological effects. (**a**) Phenolic acids and (**b**) flavonoids compounds.

**Table 1 molecules-27-02462-t001:** Compounds identified in Quince fruit (*Cydonia oblonga* Mill.) by LC-PDA-ESI-QqQ.

No.	Compound	Acronym	Retention Time (min)	Molecular Weight	Main Transition *m*/*z*	λ Max
1.	quinic acid	QA	1.31	191	93 > 85	320 nm
2.	shikimic acid	ShA	1.34	173	93 > 111	270 nm
3.	gallic acid	GaA	2.17	169	79 > 125	270 nm
4.	protocatechuic acid	PA	3.28	153	109 > 91	270 nm
5.	catechin	Cat	3.68	289	245 > 123	280 nm
6.	procyanidin B2	PB2	4.10	577	407 > 289	280 nm
7.	4-hydroxybenzoic acid	4-HA	4.24	137	93 > 65	270 nm
8.	chlorogenic acid	ChA	4.44	353	191 > 85	320 nm
9.	epicatechin	ECat	4.63	289	245 > 123	280 nm
10.	vanillic acid	VA	4.83	167	123 > 152	270 nm
11.	caffeic acid	CaA	4.92	179	135 > 89	320 nm
12.	syringic acid	SyA	5.09	197	182 > 153	270 nm
13.	2,5-dihydroxybenzoic acid	2-5-DHA	5.98	153	109 > 81	270 nm
14.	coumaric acid	CoA	6.13	163	119 > 98	320 nm
15.	rutin	R	6.41	610	609 > 300	360 nm
16.	taxifolin	Tax	6.47	125	303 > 285	290 nm
17.	ferulic acid	FA	6.70	193	178 > 134	320 nm
18.	kaempferol-3-O-glucoside	K-3-O-glu	7.28	447	255 > 284	360 nm
19.	benzoic acid	BA	7.50	121	77 > 92	270 nm
20.	2-hydroxibenzoic acid	2-HA	7.60	138	137 > 137	270 nm
21.	neohesperidin	NH	7.74	609	164 > 301	280 nm
22.	quercetin	Q	9.18	301	179 > 151	360 nm
23.	t-cinnamic acid	tCA	9.51	148	148 > 149	320 nm
24.	naringenin	N	9.81	271	151 > 119	280 nm
25.	phloretin	P	10.00	273	123 > 167	280 nm
26.	kaempferol	K	10.26	286	285 > 151	360 nm
27.	acacetin	A	12.31	269	148 > 117	320 nm

**Table 2 molecules-27-02462-t002:** Phenolic profile and content of aqueous acetone extracts obtained from Quince fruit (*Cydonia oblonga* Mill.) by LC-PDA-ESI-MS/MS.

Compound	60%	65%	70%	75%	80%	85%	90%	95%	100%
HYDROXYCINNAMIC ACIDS									
quinic acid	487.06 ± 38.34	433.33 ± 60.18	511.54 ± 27.22	562.46 ± 27.32	394.73 ± 83.38	210.55 ± 30.31	128.83 ± 12.77	59.85 ± 17.85	69.70 ± 27.82
chlorogenic acid	87.32± 6.39	99.73 ± 8.70	105.57 ± 8.61	115.14 ± 15.34	114.56 ± 17.97	120.12 ± 10.68	112.83 ± 13.84	122.02 ± 10.59	17.20 ± 4.32
caffeic acid	0.15 ± 0.02	0.16 ± 0.02	0.22 ± 0.06	0.19 ± 0.03	0.20 ± 0.01	0.23 ± 0.02	0.19 ± 0.02	0.28 ± 0.02	ND
HYDROXYBENZOIC ACIDS									
shikimic acid	9.73 ± 1.39	11.54 ± 1.34	11.58 ± 1.05	12.08 ± 1.21	15.02 ± 2.61	20.48 ± 1.13	22.70 ± 5.86	23.44 ± 2.94	3.05 ± 41.26
gallic acid	0.03 ± 0.00	0.07 ± 0.04	0.01 ± 0.00	0.03 ± 0.00	0.02 ± 0.00	0.03 ± 0.01	0.03 ± 0.01	ND	ND
protocatechuic acid	0.03 ± 0.01	0.03 ± 0.00	0.02 ± 0.01	0.04 ± 0.01	0.03 ± 0.00	0.03 ± 0.01	0.01 ± 0.00	0.02 ± 0.00	0.01 ± 0.00
2,5-dihydroxybenzoic acid	0.24 ± 0.07	0.16 ± 0.05	0.24 ± 0.08	0.33 ± 0.03	0.30 ± 0.02	0.35 ± 0.01	0.22 ± 0.01	0.23 ± 0.03	0.02 ± 0.00
vanillic acid	0.05 ± 0.02	0.03 ± 0.01	0.06 ± 0.01	0.06 ± 0.01	0.13 ± 0.04	0.06 ± 0.01	0.06 ± 0.01	0.07 ± 0.01	TR
t-cinnamic acid	0.05 ± 0.00	0.11 ± 0.01	0.07 ± 0.00	0.12 ± 0.00	0.17 ± 0.07	0.09 ± 0.00	0.13 ± 0.02	0.27 ± 0.04	0.03 ± 0.00
benzoic acid	0.61 ± 0.05	0.87 ± 0.09	0.83 ± 0.05	0.93 ± 0.2	0.94 ± 0.11	0.67 ± 0.10	0.87 ± 0.14	0.77 ± 0.21	0.68 ± 0.12
2-hydroxybenzoic acid	0.10 ± 0.00	0.10 ± 0.00	0.09 ± 0.00	0.10 ± 0.00	0.15 ± 0.02	0.09 ± 0.00	0.08 ± 0.01	0.09 ± 0.00	0.06 ± 0.03
syringic acid	TR	0.02 ± 0.00	0.03 ± 0.01	TR	0.04 ± 0.01	0.07 ± 0.02	0.07 ± 0.00	0.17 ± 0.02	ND
4-hydroxybenzoic acid	0.30 ± 0.04	0.37 ± 0.10	0.42 ± 0.03	0.45 ± 0.03	0.47 ± 0.18	0.45 ± 0.06	0.47 ± 0.10	0.50 ± 0.04	0.13 ± 0.00
coumaric acid	0.04 ± 0.01	0.07 ± 0.01	0.07 ± 0.02	0.07 ± 0.01	0.08 ± 0.02	0.08 ± 0.02	0.10 ± 0.02	0.09 ± 0.00	0.01 ± 0.00
ferulic acid	ND	0.02 ± 0.00	0.01 ± 0.00	0.02 ± 0.00	0.01 ± 0.00	0.01 ± 0.00	0.01 ± 0.00	0.03 ± 0.01	TR
FLAVAN-3-OLS									
procyanidin b2	259.59 ± 40.01	217.44 ± 45.98	184.56 ± 8.1	138.14 ± 14.1	150.29 ± 49.42	58.27 ± 19.27	28.36 ± 0.48	4.56 ± 1.72	5.24 ± 3.97
epicatechin	3.91 ± 0.60	3.27 ± 0.69	2.78 ± 0.12	2.08 ± 0.21	2.26 ± 0.74	0.87 ± 0.29	0.42 ± 0.00	0.06 ± 0.02	TR
catechin	0.19 ± 0.02	0.16 ± 0.04	0.13 ± 0.02	0.13 ± 0.02	0.11 ± 0.03	0.07 ± 0.02	0.04 ± 0.01	0.02 ± 0.00	0.02 ± 0.00
FLAVONOLS									
rutin	274.14 ± 43.18	281.96 ± 29.78	276.17 ± 61.09	232.54 ± 18.86	233.75 ± 36.43	219.77 ± 22.35	171.68 ± 19.77	214.74 ± 25.24	17.91 ± 8.68
kaempferol-3-o-glucoside	4.16 ± 0.35	4.49 ± 0.23	4.58 ± 0.19	4.08 ± 0.80	4.27 ± 0.65	4.35 ± 0.67	3.94 ± 0.65	3.93 ± 0.19	0.71 ± 0.23
quercetin	0.25 ± 0.01	0.27 ± 0.03	0.24 ± 0.08	0.34 ± 0.04	0.42 ± 0.10	0.41 ± 0.03	0.76 ± 0.00	1.03 ± 0.07	0.04 ± 0.00
kaempferol	0.02 ± 0.00	0.02 ± 0.00	0.13 ± 0.00	TR	TR	TR	TR	TR	0.17 ± 0.07
taxifolin	0.1 ± 0.00	TR	TR	TR	TR	TR	TR	TR	TR
FLAVANONES									
naringenin	ND	TR	TR	ND	TR	TR	TR	TR	ND
FLAVONES									
acacetin	TR	0.01 ± 0.00	0.01 ± 0.00	TR	TR	TR	TR	TR	TR
DIHYDROCHALCONES									
neohesperidin	0.18 ± 0.07	0.21 ± 0.07	0.23 ± 0.05	0.33 ± 0.03	0.20 ± 0.03	0.28 ± 0.07	0.25 ± 0.06	0.25 ± 0.06	0.07 ± 0.03
phloretin	TR	TR	ND	ND	TR	TR	TR	TR	ND

**Table 3 molecules-27-02462-t003:** Antioxidant activity by ORAC and DPPH assay in Quince fruit (*Cydonia oblonga* Mill.) extracts.

Acetone (%)	ORAC [(TROLOX Equivalent µM/µg)]	DPPH (RSA%) [(TROLOX Equivalent µM/µg)]
60	144.41 ± 0.77 ^a^	52.76 ± 0.63 ^a^
65	137.65 ± 0.58 ^a^	52.53 ± 0.00 ^a^
70	140.67 ± 5.43 ^a^	52.01 ± 0.83 ^a^
75	140.39 ± 1.14 ^a^	52.98 ± 0.18 ^a^
80	142.61 ± 5.68 ^a^	51.57 ± 1.86 ^a^
85	185.49 ± 2.47 ^b^	52.31 ± 0.18 ^a^
90	132.24 ± 2.02 ^c^	52.01 ± 0.63 ^a^
95	127.41 ± 0.93 ^c^	52.09 ± 0.53 ^a^
100	117.11 ± 2.43 ^d^	51.94 ± 0.45 ^a^

The literals denote the statistical difference (ANOVA, Tukey *p* < 0.001).

**Table 4 molecules-27-02462-t004:** In vitro anti-inflammatory activity of Quince fruit (*Cydonia oblonga* Mill.) extracts.

Acetone (%)	COX-2 Inhibition (%)
60	45.74 ± 0.37 ^ac^
65	40.40 ± 5.78 ^abc^
70	34.08 ± 5.05 ^ab^
75	40.87 ± 0.40 ^abc^
80	42.58 ± 2.01 ^ac^
85	52.31 ± 0.01 ^c^
90	46.94 ± 4.15 ^d^
95	15.56 ± 0.58 ^e^
100	29.19 ± 1.17 ^f^
Dup-69	80.05 ± 0.26 ^g^

DuP-697: 5-bromo-2-(4-fluorophenyl)-3-(4-(methylsulfonyl)phenyl)-thiophene. The literals denote the statistical difference (ANOVA, Tukey *p* < 0.001).

## Data Availability

Not applicable.

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
