# Peer review of "Phenolic Acids and Flavonoids in Acetonic Extract from Quince (Cydonia oblonga Mill.): Nutraceuticals with Antioxidant and Anti-Inflammatory Potential"

_molecules, 2022, doi:10.3390/molecules27082462_

Round 1

Reviewer 1 Report

The manuscript describes the optimisation of extraction solvent (% acetone) of Cydonia oblonga Mill on the yield of polyphenols as well as their antioxidant and antiinflammatory activities. 

Significant improvement is needed in a few aspects. The manuscript may benefit from proofreading by a native English speaker.

  1. The details of chemical profiling is lacking. It is unclear how the authors identified the polyphenols. How many standards were available? Please provide more details on MRM transitions. The common practice is to match the MS/MS fragmentation patterns. Please include a clearer PDA chromatogram and the TIC chromatogram.
  2. The polyphenol composition is not suitable to be presented in a bar chart as in Figure 1. The same goes for Figure 3.
  3. The positive controls for the DPPH and COX2 assays are missing. The results for the DPPH assay can be presented as Trolox eq too.
  4. In Figure 4, there isn't a clear trend between the bioactivity and the % of acetone. What is the authors' interpretation?

Reviewer 2 Report

The revision is attached in the Word document

Reviewer 3 Report

The paper submitted by Herrera-Rocha et al. is focused on the investigation of the chemical composition and bioactivity of extract prepared from fruits of quince. The paper is in general well written and the results are interesting. However, before further consideration of the manuscript, some issues should be addressed and some corrections must be introduced to the text to improve its scientific soundness and quality.

1) fid. 1 - error bars should be presented in the graph - maybe each group of phytochemicals should be presented as separate bar in the graph

2)the quality of figs and tables should be improved - figs 2, 3 - please improve the quality, tables all fonts should be the same as in the main text

3) fig 3 - please provide a new graph showing each group of phytochemicals as a separate bar with corresponding error; statistically significant differences for each group of compounds quantified should be indicated between different extracts with suitable p values

4) separation shown in fig 2 is not optimal in my opinion; the gradient is too sharp and compounds are poorly separated; it could be avoided by the modification of the gradient program; please try to re-record the sample with a gradient starting at 3% of MeCN and ending at 30 % of MeCN in 60 min; then peaks should be relabeled and retention times should be corrected 

5) the characterization of the extract is not full - what about peaks between compounds 21 and 22 - any idea on the identity?; other compounds which are not fully identified should also be reported in the manuscript

6) names of compounds in table 1 should start with small letters

7) section 2.2 more info on the authentication of the plant materials should be provided; did the authors deposit a voucher sample?; voucher specimen number should be given

Round 2

Reviewer 1 Report

The authors  stated that appropriate positive controls were used for the DPPH and COX-2 assays, however, I do not see any changes to the Results section and the relevant tables.  The discussion needs to be updated accordingly.

Reviewer 2 Report

The manuscript has been carefully revised by the Authors. Also I appreciate answers on all comments. Best wishes in your future scientific work.

Author Response

Thanks, we really appreciate attention and feedback from reviewer

Reviewer 3 Report

The authors submitted an improved version of their manuscript. Some of the points raised by me in the previous review were correctly addressed. However the paper still needs corrections to be considered for publication in Molecules.

1) in my opinion more info on the plant material used for the study must be provided; the plant material should be correctly authenticated. the methodology must be given in the manuscript; name of the person responsible for identification; identification method and citation for proper literature together with no of voucher specimens is obligatory for this kind of research

2) the authors state that the separation was improved according to my suggestion - surprisingly in the correct version of the manuscript I found the same chromatogram and the same elution gradient as in the previous version  - please be serious and provide additional experiments or prepare a reasonable rebuttal

Author Response

This manuscript is a resubmission of an earlier submission. The following is a list of the peer review reports and author responses from that submission.

Round 1

Reviewer 1 Report

After careful revision of the article by Rocha et al entitled ‘Phenolic acids and flavonoids in acetonic extract from quince (Cydonia oblonga Mill.): nutraceuticals with antioxidant and anti-inflammatory potential’ I suggest to resubmit the manuscript to a more sectorial journal due to the lack of novelty and the few experiments presented.

Reviewer 2 Report

The revision is attached in a separate document

Reviewer 3 Report

In this manuscript, the authors investigated the effect of composition of extraction solvent (aqueous acetone) on the antioxidant and antiinflammatory activities of Cydonia oblonga fruit. Standard methodologies were used and data presentation and analysis are sufficiently clear.  There are a few comments for the authors' consideration.

  1. A strong justification for the range of aqueous acetone (60-100%) should be stated.
  2. Chemical profiling of the extracts needs some improvements. The compounds that were identified by the comparison with standards should be clearly stated. Some of the peaks (#28, #29) seemed nonsignificant in the chromatogram provided. Please provide a zoomed in chromatogram for peaks like the above.
  3. Positive control for the DPPH assay is missing. Why didn't the authors use Trolox as well?
  4. An appropriate positive control for the COX-2 inhibitory assay is required.
  5. The authors are suggested to compare their findings with those who used similar combination of extraction solvents on other fruits.